# Inception Mechanisms of Tunneling Nanotubes

**DOI:** 10.3390/cells8060626

**Published:** 2019-06-21

**Authors:** Mitja Drab, David Stopar, Veronika Kralj-Iglič, Aleš Iglič

**Affiliations:** 1Laboratory of Physics, Faculty of Electrical Engineering, University of Ljubljana, 1000 Ljubljana, Slovenia; mitja.drab@fe.uni-lj.si; 2Laboratory of Clinical Biophysics, Faculty of Medicine, University of Ljubljana, 1000 Ljubljana, Slovenia; kraljiglic@gmail.com; 3Department of Food Science and Technology, Biotechnical Faculty, University of Ljubljana, 1000 Ljubljana, Slovenia; david.stopar@bf.uni-lj.si; 4Laboratory of Clinical Biophysics, Faculty of Health Sciences, University of Ljubljana, 1000 Ljubljana, Slovenia

**Keywords:** tunneling nanotubes, filopodia, anisotropic membrane domains, cytoskeletal forces, f-actin

## Abstract

Tunneling nanotubes (TNTs) are thin membranous tubes that interconnect cells, representing a novel route of cell-to-cell communication and spreading of pathogens. TNTs form between many cell types, yet their inception mechanisms remain elusive. We review in this study general concepts related to the formation and stability of membranous tubular structures with a focus on a deviatoric elasticity model of membrane nanodomains. We review experimental evidence that tubular structures initiate from local membrane bending facilitated by laterally distributed proteins or anisotropic membrane nanodomains. We further discuss the numerical results of several theoretical and simulation models of nanodomain segregation suggesting the mechanisms of TNT inception and stability. We discuss the coupling of nanodomain segregation with the action of protruding cytoskeletal forces, which are mostly provided in eukaryotic cells by the polymerization of f-actin, and review recent inception mechanisms of TNTs in relation to motor proteins.

## 1. Introduction

### 1.1. Early Experiments and Definitions

In a series of experiments in 2002 and 2003, Kralj-Iglič and colleagues observed long thin protrusions attached to giant unilamellar vesicles (GUVs), reporting on a movement of gondola-like small blebs between them (Figure 1A) [1,2]. In the final stages of such transport, the bleb was fused with the cell membrane, which releases the contents in the process (Figure 1C,D). In another experiment, similar tethers were observed in vivo in red blood cells [3]. These membrane nanotubes appeared very fragile, which render them difficult to visualize and study. However, it was proposed that the observed mechanism of nanotube-directed transport between two membrane-enclosed compartments may play an important role in live cells where it was previously overlooked.

In 2004, Rustom et al. reported on in vitro structures connecting rat kidney cells in tissue culture over long distances [4]. These structures, coined tunneling nanotubes (TNTs), were disconnected from the substrate and formed directly between neighboring cells de novo in a matter of minutes. With diameters from 50 to 200 nm and lengths that spanned up to 100 µm, the tubes displayed pronounced sensitivity to prolonged light excitation and mechanical stress. The TNTs were found to act as an inter-cellular transport system, with fluorescence video-microscopy revealing a transport of cell organelles between neighboring cells.

Soon, similar structures have been found in many prokaryotic and eukaryotic cells [3,5,6,7,8], including immune [9], urothelial [10,11] (Figure 1E–G), neuronal, and primary cells [12], where their importance in cellular communication and spreading of pathogens is becoming ever more apparent [13]. TNTs are reported to exchange mitochondrial and lysosomal organelles as well as intra-cellular vesicles, as detailed in recent reviews [13,14,15]. What defines TNTs remains an open question, since many synonyms for similar protruding structures are hindering a comprehensive definition. Recently, Dupont et al. have proposed three distinctive characteristics of TNTs: (i) they have to connect at least two cells, (ii) they do not touch the substrate, and (iii) they contain f-actin [16]. This last point segregates TNTs from any other f-actin-rich structures, such as filopodia. Perhaps a fourth requisite could be added to these three conditions. TNTs enable the transfer of cellular cargo between neighboring cells.

### 1.2. Bacterial TNTs

TNTs were also found to form between bacterial cells [17,18], which allows for the execution of antibiotic production, secretion of virulence factors, and bioluminescence [19]. Using *Bacillus subtilis* as a model organism, Dubey and colleagues found that TNTs exist both as intercellular tubes and extending tubes, with the latter frequently surrounding the cells in a “root-like” fashion, hinting at their exploring and scavenging functions. If left to grow freely, these networks become denser with time to form biofilms [20]. TNTs were found to form not only between the same bacterial species, but also among evolutionarily distant species of bacteria, hinting at a common underlying mechanism of their formation and function. When under nutrient starvation conditions, the exchange of cytoplasmic molecules was demonstrated to take place even between Gram-positive and Gram-negative anaerobes.

Nutrients, metabolites, and proteins have been demonstrated to take place under starvation between *Acinetobacter baylyi* and *Escherichia coli*. This cross-feeding phenomenon occurs in a contact-dependent fashion and relies on the formation of nanotubular structures, which implies that nanotube-associated molecular exchange is prominent in nature [21,22]. Additional membranous extracellular structures were found to serve for intercellular molecular trade as well as for other physiological properties. For example, studies in *Myxococcus xanthus* indicate the existence of intercellular outer membrane vesicle chains, encompassing outer membrane proteins known to be exchanged among cells [23,24,25]. The Gram-negative *Delftia*, on the other hand, produces membrane vesicle chains termed nanopods in response to a unique carbon source, which suggests that nanopods are involved in nutrient uptake [26]. Furthermore, nanowires in Shewanella oneidensis MR-1, utilized for electron transport, were shown to be composed of outer membrane and periplasmic extensions [27].

### 1.3. The role of TNTs in Spreading Pathogens

With emerging evidence of TNTs involvement in cancer and neurodegenerative diseases, they are becoming a promising therapeutic target [10,28,29,30]. TNTs of various morphologies have been shown to be involved in the transport of bacteria, prions, and viruses [7,31,32]. A growing amount of reports demonstrates the important role of TNTs in the pathogenesis of neurodegenerative diseases and cancer [5]. Viral transmission with TNTs was first described for the spread of human immunodeficiency virus (HIV) from infected to uninfected T cells [9,33]. This new route of HIV transmission was later confirmed in vivo within lymph nodes of mice [34]. Such hijacking of TNTs by HIV escalates viral transmission to large populations of cells and is thought to be an important factor in HIV neuropathogenesis and in the establishment of viral reservoirs [35]. Furthermore, the HIV accessory protein nef promotes actin remodeling and has been shown to stimulate the formation of TNTs [31,36].

### 1.4. TNT Growth is Dependent on Case-Specific Regulatory Proteins

There are some indications that different regulatory proteins are involved in case-specific TNT growth. One of the earliest markers discovered in the process of TNT formation was M-Sec, with its expression resulting in actin polymerization [37]. However, its expression must be reciprocated by the target cell to ensure a successful TNT formation, avoiding the creation of an incomplete close-ended TNT called a filopodial bridge [38]. A calcineurin-like protein, YmdB, was found to be required for both nanotube production and intercellular molecular trade in *Bacillus subtilis* [17]. Other recent research suggests that a filopodia-promoting network of CDC42/IRSp53/VASP in neuronal cells negatively regulated TNT formation and impaired TNT-mediated intercellular vesicle transfer. Conversely, an elevation of Eps8, which is an actin regulatory protein, increases TNT formation while inhibiting filopodia growth [30]. In PC 12 cells observed by Rustom, treatment with lantruculin-B, an inhibitor of actin polymerization, resulted in suppression of TNT growth. The polymerization of actin seems to be an almost universal trait among all eukaryotic TNT formations, but the detailed inception and the timeline of TNT production remains enigmatic.

### 1.5. A General TNT Growth-Driving Mechanism?

The aim of this paper is to review some recent TNT inception mechanisms, both from a biological and numerical modeling perspective. With such extensive diversity in morphology and structure of eukaryotic TNTs detailed in recent reviews [13,38,39], it is crucial to identify some general principles of their formation. It is clear that the cytoskeletal forces play a pivotal role in TNT formation, which arises due to localized polymerization of cytoskeleton biopolymers and due to contractile forces applied to membrane-bound filaments by molecular motors [40]. So far, it has been very difficult to study in vitro experiments of cellular membrane shape changes that involve the recruitment of the cytoskeleton [41]. Coupling of these with protrusive forces provided by the cytoskeleton such as the polymerization of f-actin may lead to a yet unknown, but possibly unifying mechanism of TNT growth.

## 2. Stability of Membranous Tubular Structures

A feature common to most TNT formations is the presence of membrane continuity [14]. Tubular membrane structures are structurally robust and are common in most cellular environments with a large surface-to-volume ratio [42]. However, continuity is difficult to assess if there is no evidence of cargo trafficking between neighboring cells.

It is widely accepted that the process of TNT formation happens in one of two ways: either the protrusion is wholly driven by polymerization of actin (type I) or cells that come into contact draw out nanotubes as they move apart (type II) [11,38] (Figure 2). Type I TNTs begin to grow like filopodia, which start to branch out as they seek connections with neighboring cells (Figure 1E–G, Figure 2). These two processes are not mutually exclusive and could occur in a unison. Many cell types store an excess of membrane in their cell structure, as shown by the ability of cells to rapidly swell when exposed to a hypertonic solution. This is best seen in dilute cultures, where a root system of nanotubes increases the cell area dramatically. A single tube of GD215 (Dhag) cells can span as much as 15 microns in 15 min, which expands the cell area by three-fold [17].

Nanotubes can be formed spontaneously in GUV systems prepared by electro-formation [43,44] (Figure 3) or can be pulled out by hydrodynamic flow, micropipettes, or optical and magnetic tweezers [45,46,47,48,49].

By using optical tweezers to pull out tubes from GUVs, Frenkel et al. investigated how area expansion affects membrane tension [51]. When a short membrane tube is elongated, there should be an observable increase in membrane tension. An increase in area by pulling out a ~10 µm of radius, 100 nm should, in principle, increase the tension by about a factor of 10 and the plateau force by about $\sqrt{10}$ [52]. However, this increase was not observed, which further supports the theory of a membrane reservoir that keeps surface tension constant. It is thought that this reservoir consists of smaller membrane inclusions, or may be due to the partial detachment of the membrane from the cover slip surface [53]. The force barrier for the formation of a membrane tube grows linearly with the size of the area, which the force is exerted on [51]. Additionally, numerical predictions show that, to form a tube, the motor proteins have to provide an initial force 13% larger than what is needed to pull a long membrane tube, which indicates that tube formation works on an all-or-nothing basis [54]. However, tubes formed from different sides of the cell result in different overshoot forces, presumably due to differences in the cytoskeletal cortex [55]. In cells, lipid subdomains of a few hundred nm size as well as clusters of proteins are found on the membrane [56,57]. A force exerted on one of these patches required for initial tube formation may be too great for force generators (e.g., kinesin, polymerizing cytoskeletal elements) to overcome [51]. Conversely, the acquisition of proteins or lipids with appropriate initial curvature may be crucial to lower the overshoot force [58].

## 3. Membrane Protrusions Coupled with Local Curvature Changes

Any adsorption of proteins or other constituents to the membrane breaks the symmetry and induces some curvature. This process introduces a local spontaneous curvature to the membrane [59]. From the analysis of such passive systems, it was found that the entropic cost prevents the aggregation of membrane inclusions in the limit of a nearly flat membrane with small height deformations, unless direct attraction between these inclusions is present [60]. The discovery of membrane-associated proteins with curved, lipid-facing surfaces (e.g. those with the BAR protein domains) has given rise to the idea that these proteins have a role in driving membrane curvature [11,50,61,62,63,64].

### 3.1. Curvature-Sensing Membrane Inclusions

There are reports where curvature-sensing proteins are observed without any *a priori* assumptions about scaffolding or attractive protein-protein interactions. In a recent study, Prévost and colleagues found the fluorescence signal of the labeled IRSp53 I-BAR protein to be very weak on the tube pulled out of the GUV with optical tweezers, which suggests that the I-BAR domain has a low affinity for positive curvature [65]. In a second experiment, the GUVs were grown in the presence of the IRSp53 I-BAR dimers, binding to both leaflets of the GUVs, and, later, transferred into a buffer detaching the proteins on the outer leaflet. When a tube was pulled from such a GUV, the protein interacted with the negatively curved inner leaflet of the nanotube results in a greatly enriched I-BAR domain on the tube. The area fraction of the proteins was around 5%. The dependence of such sorting on the tube curvature can be understood by thermodynamic arguments, which accounts for the membrane bending and stretching energies, the protein mixing entropy, and the energetic coupling between proteins and membranes [66].

### 3.2. Anisotropic Membrane Components Models

Additionally, cellular membrane constituents can be considered intrinsically anisotropic. Lipids with a large area ratio of polar head groups to acyl chains create positive curvature, while lipids with the opposite ratio create negative curvature, as displayed by the addition of cholesterol into POPC GUVs. The membrane in this model is considered to be a self-assembly of nanodomains, where the intrinsic shape of a nanodomain can be modeled within the framework of the deviatoric elasticity model with the choice of the two principal intrinsic curvatures (Figure 4A) [62,67]. Within the deviatoric elasticity model, the membrane nanodomains energetically prefer a local geometry that matches appropriate local curvature described by the two intrinsic principal curvatures (C_1m_ and C_2m_). If the two curvatures are identical, the nanodomain is isotropic, while anisotropic when they are different (Figure 4A).

The shapes of closed vesicles can be obtained numerically by direct minimization of the free energy of the membrane under constraints of constant surface area, volume, and a constant number of nanodomains [68]. It has been shown that formation of equilibrium vesicle shapes favors the accumulation of anisotropic constituents in the thin protrusive parts of the vesicles, which provide structural stability (Figure 4C). Perutkova et al. have shown that attachment of flexible rod-like proteins may minimize the total free energy of the membrane and stabilize tubular protrusions by lateral sorting [50,69]. This happens if the decrease in the orientational-free energy of the proteins is significant enough to overcome the increase of the free energy due to decreased configurational entropy [50]. Similar results have been confirmed by other studies, where the anisotropic membrane inclusions were accounted for within a deviatoric elasticity model [2,3,68,70,71,72,73].

Figure 4F shows the axisymmetric equilibrium shape of a two-component membrane with both isotropic and anisotropic components, characterized by different intrinsic curvatures. The deviatoric energy of the anisotropic component attains its minimum by accumulation of the anisotropic components in the tubular protrusion. This segregation happens due to the mismatch between the spontaneous curvature of the anisotropic component and the membrane [72]. Such formations are common in vesicular systems (Figure 4E). When the intrinsic mean curvature of the isotropic component is increased (for fixed constant volume), the length of cylindrical protrusions increases with the increase of the intrinsic mean curvature of the anisotropic component, which leads to shapes with no up-down symmetry (Figure 4D), resembling type II TNTs observed in vitro for RT4 urothelial cells (Figure 4B). The spherical vesicle at the tip of the nanotube is assumed to be stabilized by isotropic nanodomains characterized by positive (convex) curvature, while the nanotube is assumed to be formed from anisotropic cylindrical curvature-preferring membrane constituents or nanodomains (C_1m_ > 0, C_2m_ = 0). This can lead to the detachment of the tip vesicle into an extracellular vesicle [74]. Shapes with an up-down symmetry are stable only at lower values of the reduced volume [67].

### 3.3. Alternate Theories: Protein Crowding

However, there are some doubts about membrane bending exclusively by intrinsically curved proteins [77,78]. Dannhauser and Ungewickell presented in vitro evidence that clathrin alone can drive membrane deformation and that proteins such as epsin are dispensable for the creation of deeply invaginated clathrin-coated pits [79].

Another mechanism of protein crowding has been proposed recently by Stachowiak et al., by which membrane bending is driven even in the absence of any protein insertion into the lipid layer in vitro [80]. Using GUVs containing phosphatidylinositol-4,5-bisphosphate (PtdIns(4,5)P_2_) as the membrane substrate, they showed that tubulation could be achieved with either the ENTH domain of epsin or the ANTH domain of AP180 (another protein found in endocytic clathrin-coated pits and vesicles). Thus, the authors showed that high local protein concentration (crowding) adsorbed on the membrane can drive membrane curvature and formations of tubules, regardless of how the proteins are recruited on the membrane. Fluorescence-lifetime Förster resonance energy transfer (FRET) showed that tubules formed efficiently when around 20% of the membrane was covered by the epsin ENTH domain. Such crowding has yet to be demonstrated in vivo under normal conditions of protein expression.

## 4. Membrane Protrusions Coupled with Cytoskeletal Forces

There is evidence that the discussed curved membrane nanodomains are coupled with active forces arising from the cytoskeleton to deform the membrane [81,82,83]. The dynamics of membrane proteins and associated membrane deformations occur in highly non-equilibrium conditions, which means that structures not possible in equilibrium thermodynamic conditions can arise. Curved protein complexes and lipid domains respond to the shape of the membrane by aggregating where their spontaneous curvature matches the local membrane curvature. A similar mechanism of adsorption of curved molecules (such as membrane proteins) can also arise for molecules adsorbing from the cytoplasm to the membrane [84]. Proteins that are simultaneously capable of regulating actin dynamics and sensing or inducing membrane curvature could have a prominent role in TNT inception. Such modeling requires complementing thermodynamic equilibrium approaches with relevant membrane dynamics.

### 4.1. Membrane Deformation and Protrusive Forces: A Positive-Feedback Mechanism

In a model proposed by Gov and Gopinathan, such self-organization leading to TNT growth can take place via feedback mechanisms between three components: cell membrane deformation, protrusive forces of the cytoskeleton, and curved membrane activators of the cytoskeleton that correspond to the membrane shape [85]. A small membrane shape fluctuation induces a flow of curved membrane components (CMCs) towards the most protrusive part of the undulation. The flow, thus, causes an increase in local CMC concentration at the protrusive tip, which, in turn, causes the force of the cytoskeleton to increase there, pushing it further outward relative to its surroundings. As the protrusion is formed, more convex CMCs flow to the tip in a positive feedback mechanism. Protrusive forces in this model are due to actin polymerization near the membrane. Since the forces produced by the cytoskeleton arise from an energy consuming process, they cannot be derived from the Helmholtz free energy, but must be added to the equations of motion for the membrane shape [86]. This active flow of CMCs can be described by a conservation equation with included current of CMC on the membrane [86]. Without direct attraction between CMCs imposed by this current, the entropic cost prevents aggregation [60]. A simple simulation of such dynamics considers the activating membrane proteins (with own spontaneous curvature) freely diffusing in the flat membrane due to thermal fluctuations. Where the activators have high density, the actin polymerization is more extensive, which increases the normal velocity of the membrane. Such interplay between the dynamics of the membrane and the residing activators with convex spontaneous curvature predicted that the resulting membrane motion favors formation of tubular protrusions [85]. The use of continuum theory is appropriate, and, supported by in vitro experiments, even for length-scales to approximately 10 nm, which is close to the molecular size [86].

A prototypical example of this type of actin-recruiting protein with these properties is the insulin receptor tyrosine kinase substrate of 53kDa. Structural, biochemical, and cell biological experiments support the unique role of this family of proteins as transducers of signaling, which links the protruding membrane to the underlying actin cytoskeleton [87]. It has been shown that IRSp53 (Insulin Receptor Substrate of 53 kDa) can sense membrane curvature and deform the membrane [65,87]. IRSp53 is also present at low densities in the human plasma membrane at an endogenous level [88]. Its binding to VASP–the family of proteins involved in a range of processes dependent on cytoskeletal remodeling–promotes high-density clustering required for active filament elongation typical for the early stages of TNT formation in mammalian cells [89]. A recent study found that binding of Cdc42, which is a protein involved in regulation of the cell cycle, to the human Formin-like protein 1 dimer, leads to a formation of a convex surface, which assembles into an umbrella-shaped structure promoting filopodia growth [90].

Another example of such a coupled recruitment is observed with gangliosides such as GM1s, which are pentasaccharides of a strong hydrophilic character. These are docked into the outer leaflet of cell membranes. It has been shown that GM1s are aggregated at the highly curved edges of caveolae, where the large positive curvature is stabilized by hydrogen bonds between the sugar moieties of neighboring GM1s [91]. Such aggregation of GM1 molecules may generate positive membrane curvature mediating the initial growth of TNTs. The positive and negative curvatures at the outer and inner leaflets induced by a GM1 aggregate attract I-BAR domain proteins to the negative curvature at the inner leaflet of a cell membrane. The I-BAR domain proteins further bend the membrane while activating the actin nucleation machinery. The nucleation of actin filaments then drives a membrane protrusive growth, which further elongates the membrane protrusion (Figure 5A) [10]. 

### 4.2. Anisotropic Membrane Components Coupled with Protrusive Forces

Modeling actin activation forces within the deviatoric elasticity model encompasses minimizing the free energy that is composed of two main contributions: bending energy and entropic mixing of isotropic and anisotropic components [92] with the f-actin force modeled as a rigid rod constraining maximum vesicle size. The entropy of mixing enforces the isotropic and anisotropic membrane constituents to intermix. However, lateral segregation of components is not strong enough for a protrusion to form without the application of actin force. As the inner rod-like structure grows longer, the isotropic and anisotropic membrane components segregate, which facilitate the formation of a smooth tubular protrusion (Figure 5B). When the tubular protrusion gets thin enough, the anisotropic components strongly accumulate in the protrusion. This leads to a nearly complete lateral separation of isotropic and anisotropic membrane constituents. This occurs even for high values of vesicle volumes, only if actin force is applied. In the absence of a mechanical actin force provided by the rod, the equilibrium axisymmetric shapes can only form undulated (necklace-like) protrusions (Figure 6A) [93].

Similar coupling between curvature and activity was also explored theoretically outside the limit of axisymmetric shapes in a recent paper by Fošnarič et al. [94] (in press). Using Monte Carlo simulations, the vesicle membrane is represented as a triangulated, self-avoiding network, where each node can be either vacant (with zero intrinsic curvature) or occupied by an isotropic protein (with constant intrinsic curvature) that can laterally “diffuse” between nodes during each algorithm time step. The numerical equilibrium shapes without actin forces assume only protrusions with bead-like undulations, where protein domains with intrinsic curvature accumulate on the protrusions (Figure 6B). In line with experimental observations of f-actin polymerization dependency upon membrane-bound proteins, this protrusive force faces outward along the normal to the membrane at each site occupied by a protein node. The results predict the growth of tubular protrusions in the presence of cytoskeletal forces at protein concentrations of around 15% of total nodes [94].

A similar treatment can be given for a system of flexible rod-like objects attached to the closed membrane, which simulates BAR domains (Figure 7A(a)) [69]. The flexible rod-like BAR domains carry with them an orientation-dependent energy given by the angle between the principal curvature and the BAR domain (Figure 7A(b)). The orientational entropy of BAR domains is related to a rotational degree of freedom dependent on the area density of the BAR domains. At low area density, a single BAR domain is not sterically hindered by its neighbors, which renders all orientational states equally probable. Conversely, the rotation of a single domain becomes restricted at high area density due to steric interactions with neighboring domains, so the total configuration of BAR domains approaches a close-packing configuration exhibiting nematic order [69]. Such lateral orientational ordering also happens when the vesicle is elongated by an actin force from within the vesicle (Figure 7B). Laterally oriented BAR domains for a non-zero angle may form a chiral structure. As self-organization of chiral amphiphiles has been shown to promote TNT formation [96], it is possible that such a mechanism could drive TNT inception.

To summarize, anisotropic membrane components [1,93] or the local external force acting on the membrane [92] are necessary for the stability of tubular membrane protrusions growing on the surface of the parent vesicles (see Figure 5 and Figure 6). However, external local force or anisotropic membrane components are not necessary conditions for stable free tubular lipid bilayer vesicle structures at small relative vesicle volumes, at least from a theoretical standpoint (Figure 6C). Stable cylindrical vesicle shapes can be predicted theoretically at a fixed vesicle area and a fixed vesicle volume by minimization of the local isotropic membrane bending energy for small values of the area difference between the outer and inner lipid layers of the membrane bilayer (Figure 6C) [50,95,97]. The same sequence of prolated vesicle shapes seen in Figure 6C can be calculated within the spontaneous curvature model [97].

The solutions to the variational problem for determining the limiting shapes of the calculated sequence of the prolate vesicle shapes at small relative volumes (Figure 6C) are dependent on the extreme average curvature deviator and extreme average mean curvature at a given area and a volume of the vesicle [50,95,97]. The calculated limiting shapes of the sequence of the prolate vesicle shapes for small vesicle relative volumes can be necklace-shaped, corresponding to a maximal average mean curvature (right limiting shape in Figure 6C), or tubular shape with spherical caps, corresponding to a maximal possible average curvature deviator (left limiting shape in Figure 6C) [50,93,95,98]. Theoretically predicted continuous shape transition from the limiting tubular to the limiting necklace-like vesicle shape (chain of spherical vesicles connected by infinitesimally thin necks) can be driven by increasing the area difference between the outer and inner layers, where the latter is directly proportional to the average mean curvature, or by increasing the spontaneous curvature of the membrane [50,95,98].

### 4.3. Experimental Evidence of Membrane Clustering

It was shown experimentally that IRSp53 forms small clusters about one second before recruitment of VASP at the same spot, which is followed by protrusion growth at the same location a few seconds later [89]. Similar clustering was recently reported for full length MIM [99], which suggests that this may be a general feature of I-BAR domain proteins. Based on the ability of the I-BAR domain to be locally enriched and constrict a weakly curved membrane, a local and transient fluctuation of the membrane produced by the cytoskeleton could be stabilized and amplified through a phase separation into coexisting domains of different curvatures. The cluster of IRSp53 at the tip of the deformation is expected to produce a domain that helps recruit Cdc42. Furthermore, it also gathers actin-related proteins such as VASP, which eventually leads to the growth of TNTs [100]. Tubular extension follows the polymerization of actin, while the positive feedback between membrane deformation induced by actin polymerization and actin nucleator recruitment can help TNT growth [65].

Such IRSp53 clustering was also observed in vivo in live-cell imaging (U2OS cells) experiments, which proves that TNT growth is often preceded by formation of an IRSp53 cluster at the membrane. Similar clustering of I-BAR domain proteins has also been reported before filopodia elongation in mouse embryonic fibroblasts and in rat primary neurons [89,99].

## 5. Conclusions

Tunneling nanotubes appear to constitute a ubiquitous means of cellular communication and trafficking. However, the TNT field requires a unification of the terminology regarding their definition. This leaves to discussion whether f-actin is a necessary requirement for all TNTs. For example, bacterial cells appear to form TNTs without cytoskeletal formations. Crucial insights into TNT formation could be derived from new tools for detection and monitoring, such as high-speed video imaging or microfluidic systems. The main challenge remains finding molecular markers to study TNTs in vivo. There has been an important contribution of theoretical physics to the knowledge on TNTs, since their stability and possible inception mechanisms have been predicted before the experimental evidence on their existence was obtained. We expect a benefit from further development of theoretical models of biological membranes and their interactions with biological molecules.

## Figures and Tables

**Figure 1 cells-08-00626-f001:**
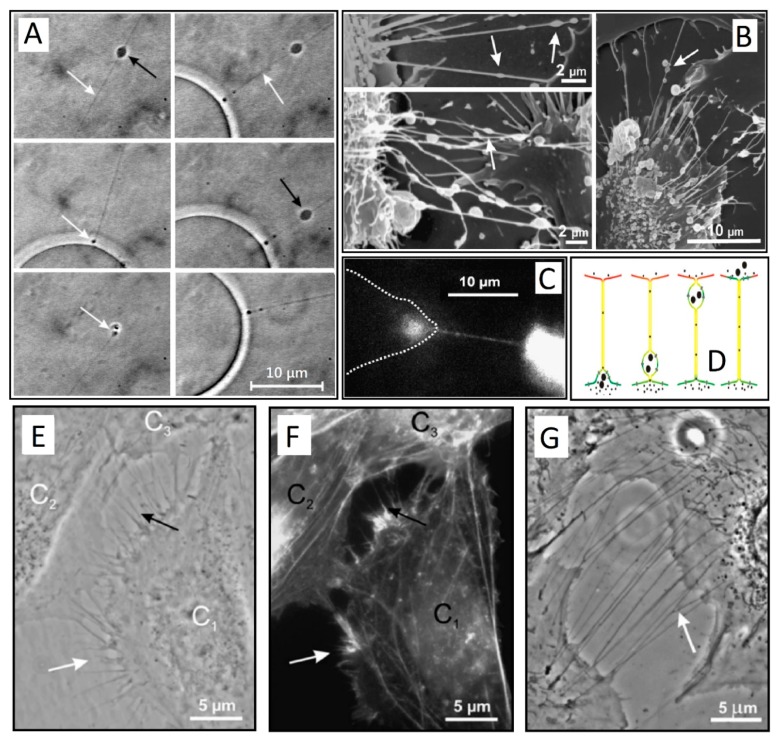
(**A**) Snapshots of a small gondola (black arrow) traveling along the thin phospholipid tube (white arrow) attached to a spherical giant unilamellar vesicle (GUV). In the final stage, the traveling gondola fused with the membrane of the GUV (white arrow) (adapted from Reference [2]). (**B**) Scanning electron microscopy of membrane nanotubes with gondolas (white arrows) observed between cells in the human urothelial cell line under physiological conditions (adapted from Reference [11]). (**C**) Exchange of actin-GFP via a TNT between two T24 cells (cell borders are indicated by a dashed line) (adapted from [11]). (**D**) A schematic illustration of TNT-directed transport between cells. Note that the gondola is the integral part of the membrane (adapted from Reference [2] and Reference [11]). (**E**,**G**) A phase contrast image of live T24 cells with type I TNTs (adapted from Reference [11]). (**F**) Fluorescence micrograph showing actin labeling of the same cells as in E after 15 min of paraformaldehyde fixation. Cell C_1_ is approaching the cells C_2_ and C_3_. The white arrows in (**E**,**F**) indicate short and dynamic membrane protrusion with which the approaching cell explores its surroundings. The black arrow in (**E**) points at protrusions that have already connected to the target cell. In all these multiple tubular connections, actin filaments are present (arrows in (**F**)). Bridging type I TNTs can be more than 20 µm in length (arrow in (**G**)) (adapted from Reference [11]).

**Figure 2 cells-08-00626-f002:**
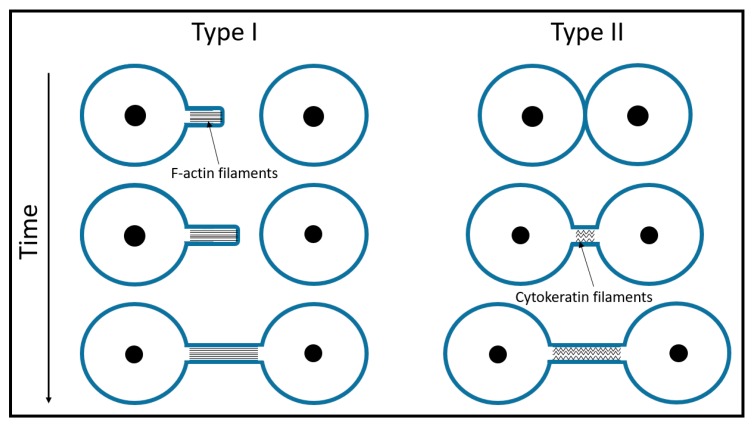
A schematic of TNT formation. Type I TNTs contain actin filaments and begin growing like filopodia. Usually, such protrusions appear in bunches of several tubes that dynamically seek connections with neighboring cells. Type II TNTs start growing as neighboring cells move apart. In the case of the urothelial lines RT4 and T24, some actin is still present at the entry point of the Type II tubes at the very beginning of the tube formation [11]. As the tube elongates, the actin gradually disappears and only cytokeratin filaments remain.

**Figure 3 cells-08-00626-f003:**
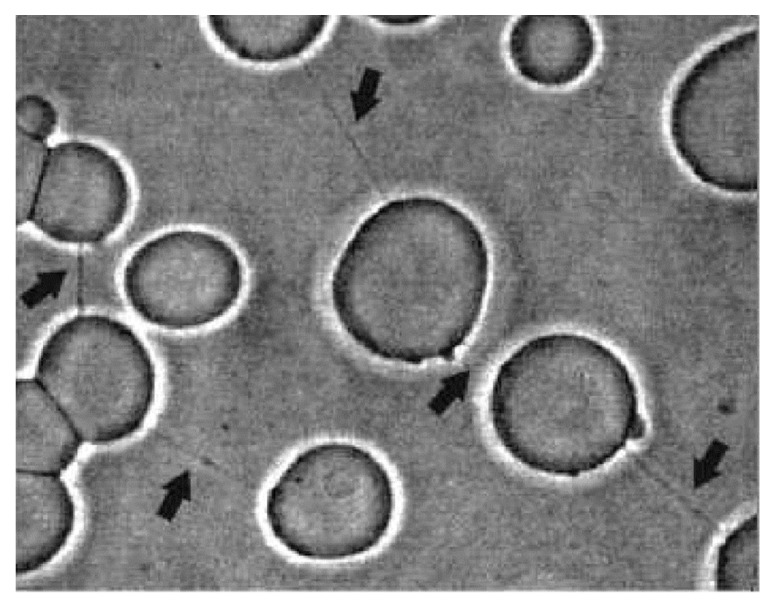
POPC–cholesterol–cardiolipin GUVs connected by thin nanotubular connections (indicated by black arrows) (adapted from Reference [50]). The GUVs were prepared by the modified method of electro-formation [43,44].

**Figure 4 cells-08-00626-f004:**
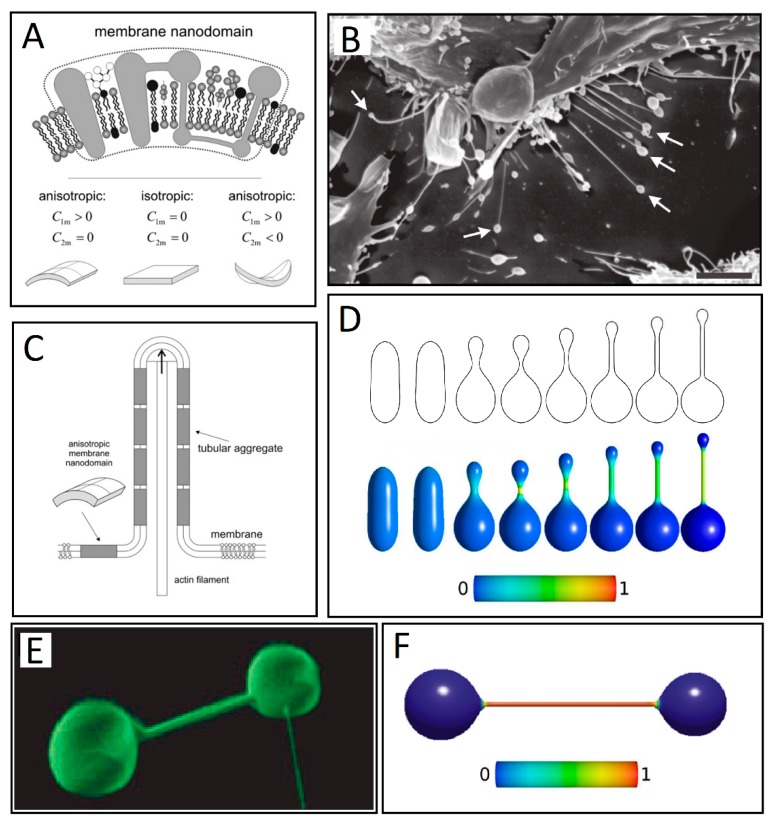
(**A**) Schematic figure of three different kinds of intrinsic shapes of flexible membrane nanodomains: partly cylindrical, flat, and saddle-like. Nanodomains with C_1m_ > 0 and C_2m_ < 0 favor saddle-like membrane geometry found in the membrane neck connecting the daughter vesicle to the parent membrane (adapted from Reference [11]). (**B**) Some of the nanotubes forming between neighboring RT4 urothelial cells have vesicles at their free tips, as indicated by the arrows. Bar = 10 μm. (adapted from [75] -under the Creative Commons 3.0 License). (**C**) Schematic illustration of stabilization of membrane protrusions by accumulation of anisotropic membrane nanodomains in the tubular region. The cylindrical-shaped anisotropic membrane domains, once assembled in the membrane region of a nanotubular membrane protrusion, keep the protrusion mechanically stable even if the cytoskeletal components (actin filaments) are disintegrated (adapted from Reference [11]). (**D**) Numerically calculated equilibrium cell shapes for a two-component membrane of isotropic and anisotropic nanodomains with constraint of constant volume. From left to right, intrinsic mean curvature of isotropic membrane components is gradually increased, which results in a more pronounced isotropic-anisotropic segregation given by the color map. The color represents the fraction area covered by isotropic components. The fully blue color corresponds to a membrane composed of isotropic nanodomains only. The shift towards green and red colors indicates increased lateral density of the anisotropic components (adapted from Reference [67] under the Creative Commons 3.0 License). (**E**) Nanotube–vesicle network stained with the membrane dye DiO (adapted from Reference [39]). (**F**) The calculated equilibrium closed membrane shape obtained in a system of isotropic components and anisotropic components drawn without the inclusion of the entropy term (adapted from Reference [76]).

**Figure 5 cells-08-00626-f005:**
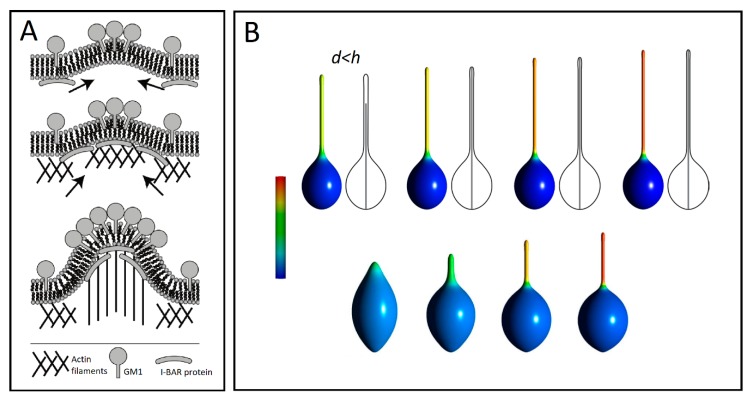
(**A**) A schematic diagram for the recruitment of I-BAR domain proteins and actin filaments by GM1 aggregates (adapted from Reference [10]). (**B**) Numerically calculated equilibrium cell shapes for different lengths *d* of the actin rod-like structure enclosed inside the vesicle of height *h*. All vesicles have the same relative volume (0.9). Apart from the top-left vesicle shape, *d* = *h*. The mixing entropy term was considered only for the second row. The color represents the fraction area covered by isotropic components. The fully blue color corresponds to a membrane composed of isotropic nanodomains only, while the yellow and red colors denote high concentrations of anisotropic components (adapted from Reference [92]).

**Figure 6 cells-08-00626-f006:**
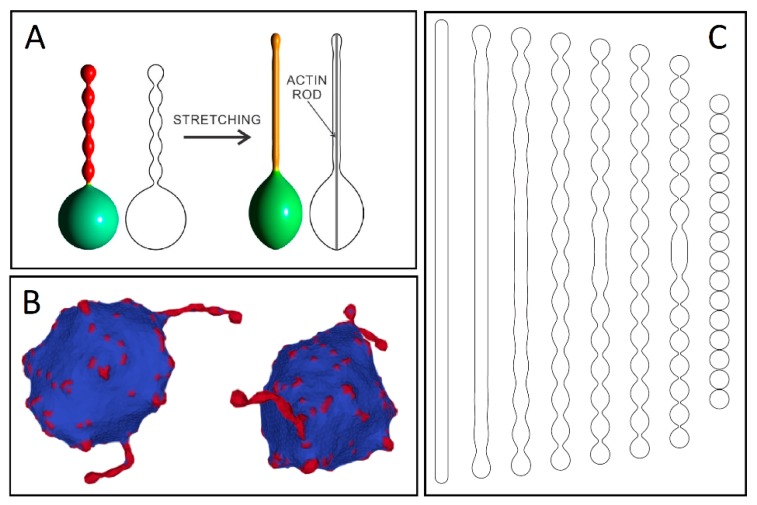
(**A**) Vesicle shapes calculated for two-component membrane, where one component has high positive intrinsic curvature. The red color represents the highest possible concentration of the membrane component with high intrinsic curvature (adapted from Reference [92]). (**B**) The results of the Monte Carlo simulation for non-axisymmetric vesicle shape without actin cytoskeleton (adapted from Reference [94]). (**C**) Analytically calculated axisymmetric vesicle shapes at small relative volume, calculated by minimization of the local membrane bending energy [95]. The calculated series of shapes is constrained by two limiting shapes, i.e. tubular vesicles on the left and necklace-like shapes on the right hand side of series of calculated shapes (adapted from Reference [50]).

**Figure 7 cells-08-00626-f007:**
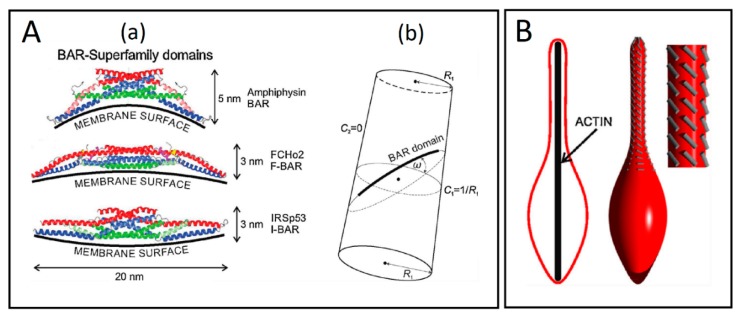
(**A**) Schematic presentations of the BAR superfamily domains and the cylindrical surface with the attached rod-like BAR domain. (**a**) The BAR domains are presented with their typical dimensions and curvature preferences. (**b**) The flexible rod-like BAR domain attached to the membrane surface of the cylindrical shape, where R1 is the radius of a cylinder. Angle ω is the angle between the normal plane of the first principal curvature C_1_ and the normal plane in which the BAR domain is lying. C_2_ is the second principal curvature (adapted from Reference [69]). (**B**) The impact of external force on the orientation of BAR domains. The lateral orientation of the membrane attached BAR domains (grey lines) is changed, when the vesicle is elongated by external force, which may result from the action of a growing cytoskeleton inside of the vesicle (adapted from Reference [69]).

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
