# Peer review of "Inception Mechanisms of Tunneling Nanotubes"

_cells, 2019, doi:10.3390/cells8060626_

Round 1

Reviewer 1 Report

Authors provide in-depth profound review on nanotubes. Not only clear EM images but simulated modeling descriptions are also highly informative. 

Author Response

(x) English language and style are fine/minor spell check required 

The spelling mistakes were corrected after another close reading of the manuscript. 

Reviewer 2 Report

In this article and colleagues review the formation and stability of tunneling nanotubes (TNTs), provided with historical background (experimental evidence) and simulation models.

The review is well written and covers recent literature about TNTs. However there are some points that authors need to revise.

1.       Before line 26 (in introduction), first define Tunneling nanotubes (TNTs). (May be first line of abstract can be repeated here).

2.       Figure 1, Figure 2 and Figure 3B are original microscopic figures adopted from other researchers. It is good authors have provided the reference (source) of each figure, please check if the journals are open access and allow the reproduction of figures or whether the permission is required from journals or authors of those figures.  

3.       While authors discuss that TNTs are the extended bridges between cells and transport materials between cells, authors should also consider that TNTs are reported to exchange mitochondrial and lysosomal organelles. Additionally, TNTs also transport intra-cellular vesicles between cells. For such information I suggest authors to refer and cite the following article (PMID: 28770210).

4.       Before discussion provide a brief paragraph with independent heading ‘’the role of TNTs in spreading pathogens’’.

5.       Line 323-324: Similar coupling between curvature and activity was also explored theoretically outside the limit of axisymmetric shapes in a recent paper by Fošnarič et al. [97] (in review). Remove the word ‘(in review)’.

Author Response

Please see attached files:

- Answers to referee (x 1)

- PDF permission forms (x 6)

Reviewer 3 Report

The review by Drab et al. focuses on the inception of tunneling nanotubes. The review provides quite a comprehensive insight into the membrane biophysical aspects of tunneling nanotube formation. It would be helpful if some further insight is provided into the cytoskeletal aspects of this process including the role of F-actin vs microtubules, the role (if any) of motor proteins both myosins, kinesin and dyneins as well as the control of actin/tubulin polymerisation. Also more on the relevance of TNTs to disease would give the review broader appeal.

Specific points:

1.     The review is quite dense in content and would be helped with further subheadings to help break up the large blocks of text and provide some more clarity.

2.     Line 83, Lantruculin B is not an enzyme

3.     Line 121, a diagram of the proposed two models of TNT formation as well as what defines a TNT should be included.

4.     Line 131, need to clarify the statement “can be pulled out”. What were the experimental techniques?

5.     Line 414, the statement that bacteria appear to form TNTs without cytoskeletal formation seems counter intuitive. Is there literature to support this statement? Bacteria certainly have homologues of actin and tubulin which could contribute to TNTs in bacteria.

6.    Line 324 and 333, what is meant by “in review”. This is listed as a 2018 paper, has it been published?

Author Response

Please see attached file:

- Answers to referee

Round 2

Reviewer 2 Report

The authors have addressed the points and have corrected their manuscript. 

It is acceptable. 

Author Response

Thank you for your acceptance.

Reviewer 3 Report

Line 96: should read "an inhibitor of actin polymerization"

Reference 106, journal name is missing.

Author Response

Thank you kindly for the comments. They were implemented.

- Line 96 now reads:

"an inhibitor of actin polymerization"

- Reference 106 on line 721 now reads:

"Fošnarič, M., S. Penič, A. Iglič, V. Kralj-Iglič, M. Drab, and N. Gov, Theoretical study of vesicle shapes driven by coupling curved proteins and active cytoskeletal forces, Soft Matter, 2019 (in print): arXiv preprint arXiv:1812.01460."